# The Dynamic Viscoelasticity of Dental Soft Polymer Material Containing Citrate Ester-Based Plasticizers

**DOI:** 10.3390/ma13225078

**Published:** 2020-11-11

**Authors:** Guang Hong, Wei-qi Wang, Lu Sun, Jian-min Han, Keiichi Sasaki

**Affiliations:** 1Division for Globalization Initiative, Liaison Center for Innovative Dentistry, Graduate School of Dentistry, Tohoku University, Sendai 980-8575, Japan; 2Department of Prosthetic Dentistry, Faculty of Dental Medicine, Airlangga University, Surabaya 60132, Indonesia; 3Division of Advanced Prosthetic Dentistry, Graduate School of Dentistry, Tohoku University, Sendai 980-8575, Japan; wangweiqi2011@yahoo.co.jp (W.-q.W.); sun.lu.e8@tohoku.ac.jp (L.S.); keiichi.sasaki.e6@tohoku.ac.jp (K.S.); 4Dental Materials Laboratory, National Engineering Laboratory for Digital and Material Technology of Stomatology, Peking University School and Hospital of Stomatology, Beijing 100081, China; hanjianmin@bjmu.edu.cn

**Keywords:** dynamic viscoelasticity, dental soft polymer, citrate ester-based plasticizer, tissue conditioner

## Abstract

The aim of this study was to investigate the dynamic viscoelasticity of dental soft polymer material containing citrate ester-based plasticizers. Three kinds of citrate ester-based plasticizer (Citroflex^®^ C-2: TEC, Citroflex^®^ A-2: ATEC, and Citroflex^®^ A-4: ATBC), with the combination of 5 wt% ethyl alcohol, were used as the liquid phase. The dynamic viscoelastic properties of nine ethyl methacrylate polymers: (A, B, C, D, E, F, G, H, and I) were immersed in 37 °C distilled water for 0, 1, 3, 7, 14 and 30 days, respectively. The dynamic viscoelastic properties were measured at 37 °C with an automatic dynamic mechanical analyzer. The shear storage modulus (*G′*), shear loss modulus (*G*″), and loss tangent (tan *δ*) were determined at 1 Hz. These parameters were statistically analyzed by two-way and one-way ANOVA and Tukey’s multiple comparison test at a predetermined significance level of 0.05. A significant difference was found among the materials in terms of the dynamic viscoelasticity. The materials containing citrate ester-based plasticizer ATBC showed the most stable dynamic viscoelasticity. Considering the limitations of this study, the results suggest that the inclusion of citrate ester-based plasticizer can improve the durability of dental soft polymer materials.

## 1. Introduction

Soft polymer materials have specific and unique viscoelastic properties and are often used in the dental field as resilient denture liners and tissue conditioners [1,2,3]. In this context, tissue conditioners are widely used for the treatment of alveolar mucosal lesions, such as decubitus ulcers. They may also be used for temporary relining during the healing phase after implant placement and other clinical applications [4,5]. As mentioned above, since the dental soft polymer material directly contacts the oral mucosa or the wound in the mucosa, the biological safety is one of the most important factors for this kind of material.

In recent years, various types of soft polymer materials for dental use that can be applied as tissue conditioners have been developed and widely used in clinical prosthetics as a pre-treatment to restore the alveolar mucosa to its normal state when a denture wearer has lesions of the alveolar mucosa [2]. The composition of this kind of dental soft polymer consists of a powder component and liquid component [6,7]. The powder component is mainly poly-ethyl methacrylate (PEMA) and its related copolymers and the liquid component is a mixture of plasticizer and ethyl alcohol [6,8]. However, during the clinical use of commercially dental soft polymer materials, the materials rapidly lose their viscoelasticity over time due to plasticizers and other substances leaching into the oral environment and water adsorption [8,9,10,11,12]. The leached plasticizer can cause the denture base material to deteriorate because the leached plasticizer will plasticize the acrylic denture base resin materials [13]. This is one of the problems that should be solved when developing new dental soft polymer materials [1,5,14]. In addition, phthalate esters, which are the most widely used plasticizers for commercially dental soft polymer, have recently been reported to exhibit estrogenic effects in various in vitro and in vivo studies and have been attracting attention as environmental hormones [15,16,17,18]. Furthermore, phthalate esters in various dental materials as plasticizers have been founded to exhibit estrogenic activity in vitro [17]. Recently, the European Union issued several directives restricting the use of three of the most common phthalates (bis (2-ethyl hexyl) phthalate, dibutyl phthalate, and benzyl butyl phthalate) in toys and childcare products and warning of their dangers [19]. Therefore, there is an urgent need to develop dental soft polymer materials, such as tissue conditioners with an improved durability, using safe plasticizer components without estrogenic activity.

Citroflex^®^ is a highly safe plasticizer made by citric acid developed by Pfizer Inc. in the New York, NY, United States [20,21]. Citroflex^®^ is a non-toxic plasticizer approved by the Food and Drug Administration (FDA) and is compatible with a variety of useful polymers, including acrylic, methacrylic, ethyl cellulose, and vinyl acetate, and is especially recommended for applications where non-toxicity is a primary requirement, such as food packaging wrap and cosmetics [22,23]. In addition, this citrate ester has a larger molecular weight than phthalate esters and is therefore less likely to leach into the environment where it is used, making it a plasticizer that is not only safe, but also one that can be expected to improve the durability of materials.

Our study aimed to develop a safe, stable, and long-lasting dental soft polymer material using citrate ester as a plasticizer by examining the dynamic viscoelasticity of the combination of citrate ester-based plasticizer and polymers. In this study, we used three kinds of citrate ester-based plasticizer and nine kinds of ethyl methacrylate polymer to investigate the effects of each combination on the dynamic viscoelastic properties of the soft polymer materials and their durability. We hypothesized that the citrate ester-based plasticizer can improve the durability of dental soft polymer materials and can be used as an alternative to phthalate ester-based plasticizers.

## 2. Materials and Methods

### 2.1. Sample Preparation

Nine poly (ethyl methacrylate) (PEMA) powders (Negami Chemical Industrial Co., LTD., Ishikawa, Japan) with different molecular weights and different particle sizes were used in powder form (Table 1). Three citrate ester-based plasticizers (Tokyo Chemical Industry Co., LTD., Tokyo, Japan), consisting of Citroflex^®^ C-2 [Triethyl Citrate (TEC)], Citroflex^®^ A-2 [Acetyl Triethyl Citrate (ATEC)], and Citroflex^®^ A-4 [Acetyl Tributyl Citrate (ATBC)], were used in liquid form (Table 2).

Table 3 shows the 27 formulations employed in this study and the powder/liquid (P/L) ratio. The powder/liquid mixture (obtained via mixing by hand for 2 min at 23 ± 2 °C and 70% humidity) was poured into polypropylene containers, glass plates were placed on top of the containers, and the samples were stored in a 37 °C incubator for 30 min. Then, the specimens were immersed in 100 mL distilled water in dark brown containers at 37 °C.

### 2.2. Dynamic Viscoelasticity Measurement

After 1, 3, 7, 14, and 30 days of immersion in distilled water, the dynamic viscoelasticity in the shear mode at 37 °C was measured five times for each formulation using an automatic dynamic mechanical analyzer (DMA Q800, TA Instruments Co., New Castle, DE, USA) (Figure 1). The specimens before immersion were used as the control. The device employed consists of a measurement operation block, furnace, and sample clamps, and is based on the principle of non-resonance-forced vibration (Figure 1). Dynamic shear displacement was applied to the specimen, and the shear storage modulus (*G′*), shear loss modulus (*G″*), and loss tangent (tan *δ*) were determined over a frequency range of 0.05–100 Hz with a 0.08% strain [5].

From a rheological point of view, the dynamic viscoelasticity was analyzed by measuring the resulting displacement (strain) when an oscillatory force (stress) was applied to a soft polymer material. When stress and strain occur in the same phase, it is a purely elastic material (a), and when the strain occurs with a 90° (π/2 radian) phase delay for the stress, it is a purely viscous material (b). Viscoelastic materials (c) display a phase delay in strain and exhibit behavior intermediate between purely viscous and purely elastic materials (Figure 2).

### 2.3. Statistical Analysis

All data on the dynamic viscoelasticity (*G′*, *G″*, and tan *δ*) were analyzed independently by one-way analysis of variance (ANOVA), and the differences among materials and among immersion times were tested with a Tukey’s multiple comparison test at a 5% level of significance. Two-way ANOVA was used to determine the influence of the composition and immersion time on the dynamic viscoelasticity of the materials. A frequency of 1 Hz was selected for statistical analyses. All analyses were computed with SPSS (SPSS 12, SPSS Japan Inc., Tokyo, Japan).

## 3. Results

The mean and standard deviations (S.D.) of the shear storage modulus (*G′*), shear loss modulus (*G″*), and loss tangent (tan *δ*) with 1 Hz before the immersion in distilled water (control group) are shown in Figure 3, Figure 4 and Figure 5. Significant differences were found among the different materials (*p* < 0.05, one-way ANOVA). The PEC-A group exhibited lower values of *G′* (*p* < 0.05, Tukey’s test) than the other formulations. The specimens containing TEC plasticizers displayed lower values of *G″* than other materials, except for the formulations containing PEMA-G, PEMA-H, and PEMA-I (*p* < 0.05, Tukey’s test). The highest values of tan *δ* were found for the group using PEMA-A in the powder form (*p* < 0.05, Tukey’s test). The materials employed in this study showed higher values of *G′* and *G″*, and similar values of tan *δ*, compared with materials that used conventional phthalate ester as a plasticizer measured in our previous study [3].

The variations of *G′*, *G″*, and tan *δ* values with the time of immersion at 1 Hz are shown in Figure 6. These rheological parameters changed over time and tended to significantly increase in value with time, except for the group using ATBC in the liquid form (*p* < 0.05, ANOVA). The group using ATBC in the liquid form exhibited a different trend, with the formulation of PEMA-A, PEMA-B, PEMA-C, and PEMA-C in the powder form showing more stable values of *G′* and *G″* over time than the other formulations. For the tan *δ*, the formulation of PEMA-A+ATBC showed the most stable value over time when compared to the other formulations. This group also demonstrated a more stable tendency in terms of viscoelastic parameters than conventional phthalate ester-based materials measured in our previous study [3].

Figure 7 shows the frequency dependence of *G′*, *G″*, and tan *δ* for the control group of each formulation. All specimens exhibited higher values of *G′* and *G″* at higher frequencies. This tendency was more remarkable for the *G″*. In all specimens, the tan δ decreased with an increasing frequency from 0.05 to 1 Hz, and then increased again at higher frequencies.

## 4. Discussion

Soft polymer materials are widely used in various fields, such as in the dental field as resilient denture liner and tissue conditioner, and so on [1]. However, there are still issues that need to be solved, such as the durability of the material, which degrades due to plasticizer leaching and water absorption, and safety issues caused by the estrogenic activity of phthalate ester plasticizer [1,3,8,15,17]. The present study investigated the effect of a combination of ethyl methacrylate polymer (PEMA) and citrate ester-based plasticizer on the dynamic viscoelastic properties of the soft polymer materials and their durability. Our hypothesis that citrate ester-based plasticizer can improve the durability of dental soft polymer materials and can be used as an alternative to phthalate ester-based plasticizers could be partially accepted. Although the durability of the dynamic viscoelastic properties of soft polymer materials was improved, some formulations were found to degrade over time, suggesting that a judicious choice of components and formulation is necessary.

The viscoelasticity of dental soft polymer material has been reported by static tests, such as stress relaxation tests, creep tests, and dynamic viscoelasticity measurements using the non-resonance-forced vibration method [2,3,6]. In general, dynamic tests are superior to static tests for very short-time behavior. Considering the fact that dental soft polymers, especially tissue conditioners, are subjected to instantaneous forces, such as chewing, the analysis of viscoelasticity by dynamic testing is very useful [6]. The chewing cycle in the oral cavity is generally 1 Hz. Therefore, the behavior of viscoelastic characteristics under typical chewing conditions can be simulated at this frequency. This is the reason why the dynamic viscoelastic parameters at 1 Hz were selected for evaluation in this study [6,24]. From the standpoint of rheology, for viscoelastic materials, the stored energy is measured by the shear storage modulus (*G′*), which represents the elastic portion, and the energy dissipated as heat is measured by the shear loss modulus (*G″*), which represents the viscous portion. The loss tangent (tan δ) is a useful parameter for measuring the ratio of energy dissipated to energy stored. Dental soft polymer materials are often employed in edentulous patients to treat alveolar mucosa lesions caused by ill-fitting dentures utilizing their specific viscoelastic properties [2]. Therefore, viscoelastic materials with higher viscous elements than elastic elements are more effective in relieving functional stress and are suitable for dental soft polymer, especially for tissue conditioner. This is because the lower storage modulus (elastic elements) and higher loss modulus (viscous elements) mean that the material has good flow characteristics and a high loss tangent. From this point of view, the group using PEMA-A in the powder form in this study showed lower *G′* and *G″* values and the highest values of tan *δ*. Therefore, it is considered to be the most appropriate dental soft polymer material.

Differences in dynamic viscoelastic behavior were observed among the different formulations. The formulations of PEMA-A+ATBC, PEMA-B+ATBC, PEMA-C+ATBC, and PEMA-C+ATBC exhibited more stable values of *G′* and *G″* over time than the other formulations. For the tan *δ*, the formulation of PEMA-A+ATBC displayed the most stable value over time in comparison to the other formulations. For the dental soft polymer, materials with a large initial flow and a lower change rate of flow over time are suitable for tissue conditioning [3,6]. This means that PEMA-A+ATBC is probably suitable for tissue conditioning. The group using PEMA-A, -B, -C, and -D as the powder form and the group using PEMA-E, -F, -G, -H, and -I as the powder form showed different tendencies in terms of the initial *G′* and *G″*. The molecular weight has a significant effect on the viscoelasticity of polymeric materials [7]. The PEMA-A, -B, -C, and -D used in this study had the same particle size (around 20 μm), but different molecular weights (around 1.0 × 10^5^, 2.0 × 10^5^, 3.0 × 10^5^, and 5.0 × 10^5^, respectively), while the PEMA-E, -F, -G, -H, and -I had the same molecular weight (around 5.0 × 10^5^), but different particle sizes (around 220, 170, 100, 60, and 40 μm, respectively). This may be the reason for the different tendencies recorded.

The largest change of *G′* and *G″* was observed in the group using TEC as the plasticizer, followed by the group using ATEC as the plasticizer. The group using ATBC as the plasticizer showed the most stability, coupled with a low change rate of *G′* and *G″*. When the polymer powder and plasticizer of soft polymer material are mixed, the polymer dissolves in the plasticizer and polymer chain entanglement occurs, finally forming a gel. These polymer dissolution behaviors are affected by the solubility parameter (SP). Generally, polymers dissolve in specific solvents that have the same SP. Even though the SP of PEMA is around 9–10, and the SP of Citroflex^®^ is also about 9, there are a few differences among the TEC, ATEC, and ATBC. These differences can affect the viscoelastic properties of the materials. Therefore, it is assumed that they show differences in viscoelasticity tendencies in different combinations. The gel formed by dissolution of the polymer is not cross-linked, and only exhibits polymer chain entanglements. However, the plasticizer can easily leach out of the material because it only intervenes between the polymer chains and does not undergo any chemical reaction. Many researchers have reported that the change in the viscoelasticity of the soft polymer materials during use is due to the leaching out of the plasticizer from the materials [1,3,9,10]. Jones et al. [8] reported that plasticizers with higher molecular weights leached less from soft polymer materials than plasticizers with a lower molecular weight. In our previous study, we also reported that the phthalate ester-based plasticizer easily leached out into immersion solutions and recorded rapid changes in viscoelastic properties [3]. The molecular weights of plasticizers used in this study are 276.28 (TEC), 318.32 (ATEC), and 402.48 (ATBC). In particular, the molecular weight of ATBC is also higher compared to conventional phthalate ester-based plasticizers such as dibutyl phthalate (278.35), butyl phthalyl butyl glycolate (336.38), and benzyl butyl phthalate (2312.36), etc. This is considered to be the reason why groups using the ATBC as the plasticizer show the best durability of viscoelastic properties, except for the formulations using PEMA-D, PEMA-E, PEMA-F, PEMA-G, and PEMA-I as the powder form. Murata et al. [25] reported that the most suitable combination of components can provide the best conditions for the optimal formation of pseudo cross-links consisting of polymer chain entanglements. This may be the reason why the above phenomenon occurs.

The viscoelastic parameters of all specimens showed sensitivity to the changes in frequency before the immersion in distilled water. The *G′* and *G″* exhibited higher values at higher frequencies. The tan δ of all specimens decreased with an increasing frequency from 0.05 to 1 Hz, and then increased again from 1 to 100 Hz. This means that the viscoelastic properties of the dental soft polymer material will vary, depending on the mastication frequency. When chewing quickly, the viscous element of the material becomes more pronounced than the elastic element. 

Polymeric materials used in dentistry not only require special mechanical properties such as viscoelasticity, but must also be antimicrobial, biocompatible, and non-cytotoxic [26,27]. The citrate ester-based plasticizer (Citroflex^®^), especially ATBC, is a highly safe plasticizer and is a non-toxic plasticizer approved by the Food and Drug Administration (FDA). This plasticizer is especially recommended for applications where non-toxicity is a primary requirement [21,23]. Therefore, in this study, we used Citroflex^®^ as a plasticizer. In addition, denture stomatitis is often observed in elderly people, where soft polymer materials are commonly used, and is a major problem. For this reason, anti-fungal properties are one of the most important factors when developing new dental soft polymer materials [28,29,30,31,32]. However, further study on the antimicrobial and antifungal properties is needed.

The results of this study indicate that the dynamic viscoelasticity of dental soft polymer materials was greatly influenced by the combination of plasticizer and polymer powder. The formulation using ATBC as the liquid form and PEMA-A, PEMA-B, PEMA-C, and PEMA-C as the powder form showed a stable tendency of dynamic viscoelasticity. However, the present study did not fully simulate clinical behavior because all specimens were immersed in distilled water and measured in a dry state. To overcome the limitations of these in vitro tests, artificial saliva should be used as an immersion solution, and the leachability of plasticizer, water sorption and solubility, and gelation time need to be investigated in order to understand how different components affect the durability and viscoelasticity of dental soft polymer material.

## 5. Conclusions

From the standpoint of dynamic viscoelasticity, Citroflex^®^ A-4 (ATBC) is the most suitable plasticizer for dental soft polymer material. Considering the limitations of this in vitro study, the results suggest that the inclusion of citrate ester-based plasticizer can improve the durability of dental soft polymer materials.

## Figures and Tables

**Figure 1 materials-13-05078-f001:**
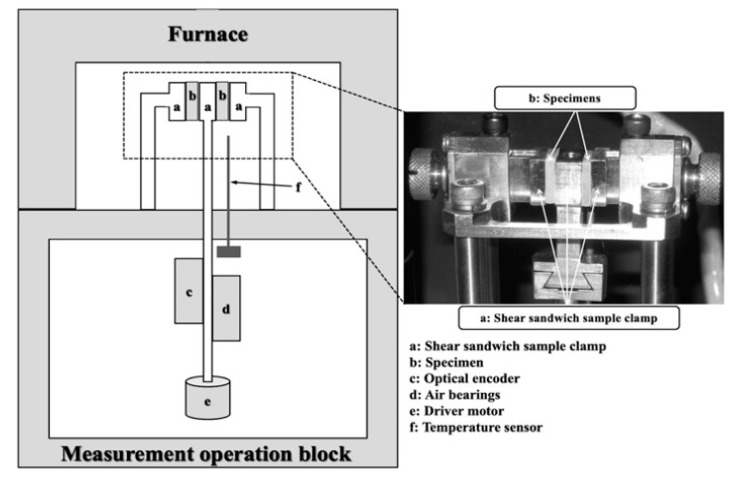
Block diagram of the automatic dynamic mechanical analyzer and shear sandwich sample clamp.

**Figure 2 materials-13-05078-f002:**
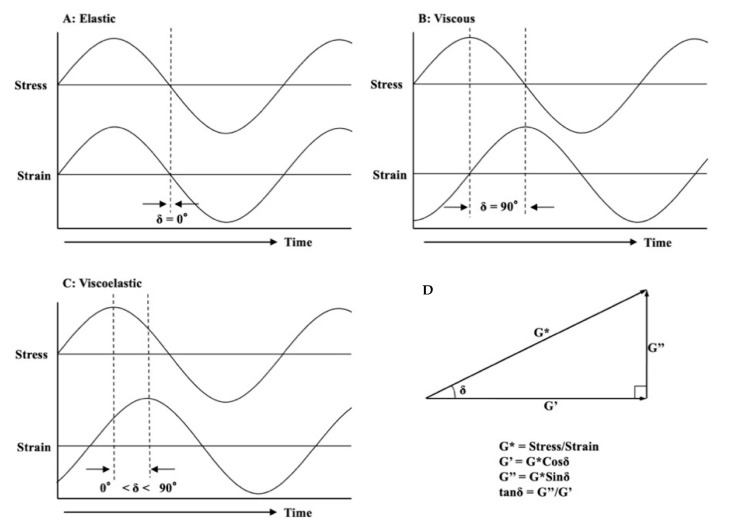
Schematic representation of the relationship between stress and strain. (**A**): Elastic behavior; (**B**): Viscous behavior; (**C**): Viscoelastic behavior; (**D**): Calculation of viscoelastic parameters.

**Figure 3 materials-13-05078-f003:**
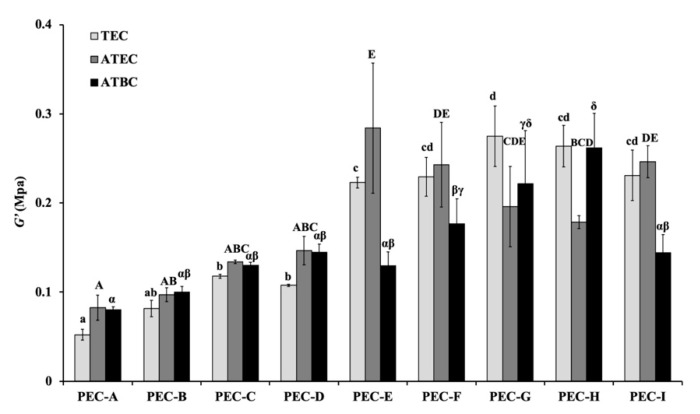
The storage modulus (*G*′) values at 1 Hz for materials before immersion. Identical letters indicate no significant differences (*p* > 0.05).

**Figure 4 materials-13-05078-f004:**
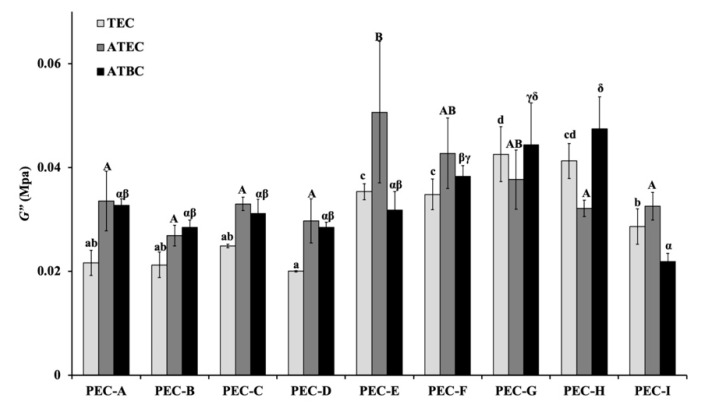
The loss modulus (*G″*) values at 1 Hz for materials before immersion. Identical letters indicate no significant differences (*p* > 0.05).

**Figure 5 materials-13-05078-f005:**
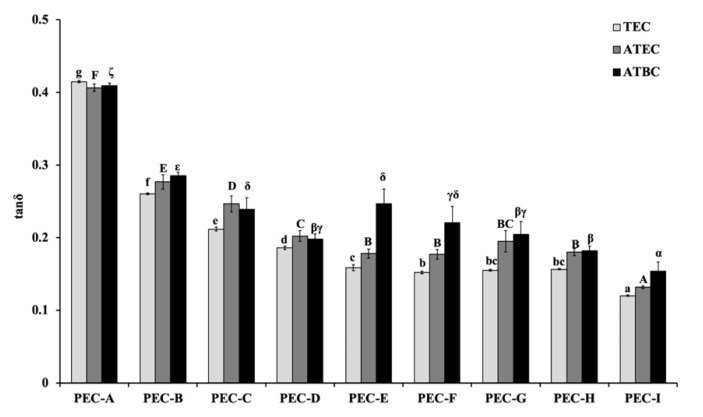
The loss tangent (tan *δ*) values at 1 Hz for materials before immersion. Identical letters indicate no significant differences (*p* > 0.05).

**Figure 6 materials-13-05078-f006:**
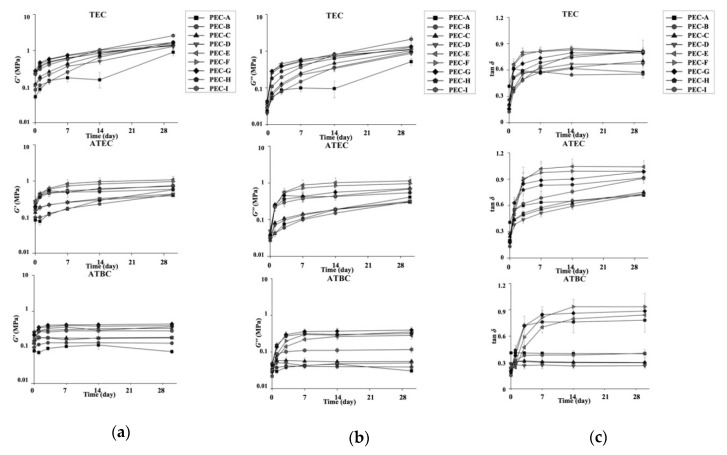
Variations of viscoelastic parameters with time of immersion at 1 Hz. (**a**) Storage modulus (*G′*) values; (**b**) loss modulus (*G″*) values; and (**c**) loss tangent (tan *δ*) values.

**Figure 7 materials-13-05078-f007:**
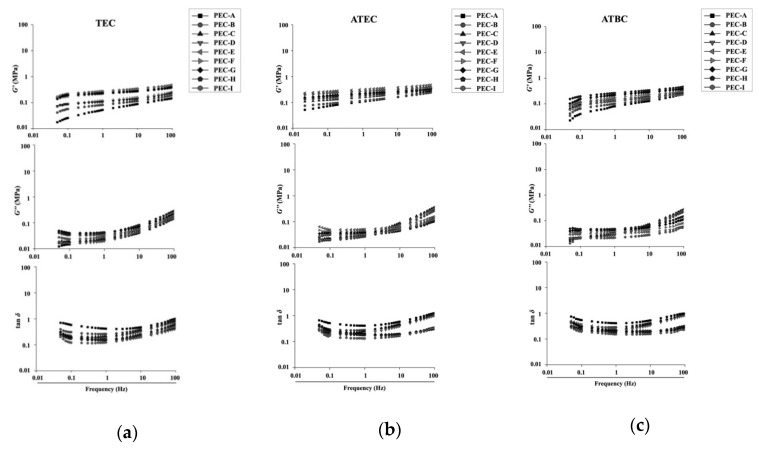
Variation of the storage modulus (*G′*), loss modulus (*G″*), and loss tangent (tan *δ*) with frequency for each group before immersion: (**a**) Triethyl Citrate (TEC) group; (**b**) Acetyl Triethyl Citrate (ATEC) group; and (**c**) Acetyl Tributyl Citrate (ATBC) group.

**Table 1 materials-13-05078-t001:** Polymer powders used.

Code	Polymer	Weight-Average Molecular Weight	Particle Size (μm)	Manufacturer
PEMA-A	SDP-32A	1.08 × 10^5^	23.3	Negami Chemical Industrial Co., LTD., Ishikawa, Japan
PEMA-B	SDP-32B	2.39 × 10^5^	18.6	
PEMA-C	SDP-32C	3.75 × 10^5^	19.1	
PEMA-D	SDP-32D	5.30 × 10^5^	18.6	
PEMA-E	SDP-31E	5.15 × 10^5^	221.3	
PEMA-F	SDP-31F	5.14 × 10^5^	176.5	
PEMA-G	SDP-31G	5.34 × 10^5^	100.2	
PEMA-H	SDP-31H	5.36 × 10^5^	63.1	
PEMA-I	D-250E	5.00 × 10^5^	37.5	

PEMA: Poly (ethyl methacrylate).

**Table 2 materials-13-05078-t002:** Plasticizers used.

Plasticizer	Manufacturer	Lot. No.
Citroflex C-2 [Triethyl Citrate (TEC)]	Tokyo Chemical Industry Co., LTD., Tokyo, Japan	A0822
Citroflex A-2 [Acetyl Triethyl Citrate (ATEC)]	Tokyo Chemical Industry Co., LTD., Tokyo, Japan	A0068
Citroflex A-4 [Acetyl Tributyl Citrate (ATBC)]	Tokyo Chemical Industry Co., LTD., Tokyo, Japan	C0367
Ethyl alcohol (EtOH)	Wako Pure Chemical Industries Ltd., Osaka, Japan	KWP4183

**Table 3 materials-13-05078-t003:** Formulations of components.

Code	Powders	Liquids	P/L by Weight
PEC-A	PEMA-A 100wt%	TEC or ATEC or ATBC 95wt% + EtOH 5wt%	1.35
PEC-B	PEMA-B 100wt%		
PEC-C	PEMA-C 100wt%		
PEC-D	PEMA-D 100wt%		
PEC-E	PEMA-E 100wt%		
PEC-F	PEMA-F 100wt%		
PEC-G	PEMA-G 100wt%		
PEC-H	PEMA-H 100wt%		
PEC-I	PEMA-I 100wt%		

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
