# Peer review of "The Dynamic Viscoelasticity of Dental Soft Polymer Material Containing Citrate Ester-Based Plasticizers"

_materials, 2020, doi:10.3390/ma13225078_

Round 1
Reviewer 1 Report
Dear Authors,
I carefully reviewed the manuscript materials-987605, the topic is quite interesting and you afford a relevant issue. Nevertheless, there are some points that in my opinion need some clarification.
In material preparation you mentioned a dispersion by hand for 2 minutes. Since this kind of dispersion is not very efficient, did you do some experimental verification about the dispersion level and homogeneity of the powder in the liquid? Is the dispersion good? In fact the rheological properties are quite affected by the dispersion level, therefore this verification is fundamental for the reliability of the viscoelasticity data, and to confirm all analysis you did on rheological curves and G’, G’’ tan_delta values.
The results and the relative discussion could be improved including the properties of the standard phthalate esters based material as reference. In this way the improvements you mentioned are clearly visible.
Best Regards
Author Response
Response to Reviewer 1 Comments
We are grateful to Reviewer 1 for the critical comments and useful suggestions that have helped us improve our paper. As indicated in the responses that follow, we took all these comments and suggestions into account when revising our manuscript. Please note that in our revised manuscript, all revisions are red-highlighted.
Comments from Reviewer 1
I carefully reviewed the manuscript materials-987605, the topic is quite interesting and you afford a relevant issue. Nevertheless, there are some points that in my opinion need some clarification.
Point 1: In material preparation you mentioned a dispersion by hand for 2 minutes. Since this kind of dispersion is not very efficient, did you do some experimental verification about the dispersion level and homogeneity of the powder in the liquid? Is the dispersion good? In fact the rheological properties are quite affected by the dispersion level, therefore this verification is fundamental for the reliability of the viscoelasticity data, and to confirm all analysis you did on rheological curves and G’, G’’ tan delta values.
Response 1: Thank you for your very important comment. As you know, when these kinds of materials are used in clinical practice, they are usually mixed by hand. Therefore, the clinical situation should be simulated as much as possible. Also, several preliminary experiments were conducted before the main experiment. In preliminary experiments, the different powder/liquid ratios and different alcohol contents were used to make the most stable and well-dispersed mixture of powder and liquid, and finally, we determined the best powder/liquid ratio and alcohol content for use in the main experiment. Therefore, we used hand mixing in this study. And also all specimens dispersed well and stable.
Point 2: The results and the relative discussion could be improved including the properties of the standard phthalate esters based material as reference. In this way the improvements you mentioned are clearly visible.
Response 2: Thank you for your very useful comment. Actually, we already down the experiment regarding the dynamic viscoelasticity of standard phthalate ester-based materials, and published in an international journal (Gerodontology, Ref. 3). Based on your comment, I added several sentences to the Results (Line 132-134, 150-152) and Discussion (Line 226-228, 229-231) section, to compare the properties between phthalate ester-based material and citrate ester-based materials.

Reviewer 2 Report
" The dynamic viscoelasticity of dental soft polymer material containing citrate eater based plasticizers.”
It is very interesting to examine the dynamic viscoelasticity of the material by making a trial material using a combination of citrate ester-based plasticizer and polymers. This paper is very well drawn. However, there are some corrections that are essential to meet the standard for publication. Please refer to the following comments.
1) There are many plasticizers in the world. Please briefly add the motivation for why we chose these plasticizers.
2) The authors consider that the difference in dynamic viscoelastic behavior is related to the difference in molecular weight and the difference in particle size of components. However, the range of difference in dynamic viscoelastic behavior of each component varies depending on each plasticizers. Please add to the discussion about the cause of this.
3) Many of the cited references used in this manuscript are old. Please also add the latest findings.
Author Response
Response to Reviewer 2 Comments
We are grateful to Reviewer 2 for the critical comments and useful suggestions that have helped us improve our paper. As indicated in the responses that follow, we took all these comments and suggestions into account when revising our manuscript. Please note that in our revised manuscript, all revisions are red-highlighted.
Comments from Reviewer 2
It is very interesting to examine the dynamic viscoelasticity of the material by making a trial material using a combination of citrate ester-based plasticizer and polymers. This paper is very well drawn. However, there are some corrections that are essential to meet the standard for publication. Please refer to the following comments.
Point 1: There are many plasticizers in the world. Please briefly add the motivation for why we chose these plasticizers.
Response 1: Thank you for your comment. I added the sentence in the Discussion section (Line 245-249) to indicate the reason why we chose the Citroflex®️ as a plasticizer. Also, in Line 62-69, the advantage of Citroflex®️ was listed.
Point 2: The authors consider that the difference in dynamic viscoelastic behavior is related to the difference in molecular weight and the difference in particle size of components. However, the range of difference in dynamic viscoelastic behavior of each component varies depending on each plasticizers. Please add to the discussion about the cause of this.
Response 2: Thank you for your comment. I added the discussion in Line 215-220 as your suggestions.
Point 3: Many of the cited references used in this manuscript are old. Please also add the latest findings.
Response 3: Thank you for your comment. I added several articles published within 5 years.

Reviewer 3 Report
INTRODUCTION
Line 35: actually, I don’t find “to dental filed” very clear. Did you mean “in the dental field”?
Line 38: “they are may also be used” should be rephrased as “they may also be used”
Line 40-41: “is the one of important factor” should be rephrased as “is one of the important factors”
Line 44: substitute “have” with “has”
Line 70: substitute “materials” with “material”
Line 71-72: I think the sentence should be rephrased properly
MATERIALS AND METHODS
Line 86: how centigrade and humidity were assessed?
Line 102: substitute “The specimens before immersion were used as controls” instead of “The specimens
before immersion was used as a control”
In the section of materials and methods you should add references as regards the methodology used
DISCUSSION
I suggest you to add in your discussions that many efforts should be done in dentistry not only to propose materials with anti-infective characteristics (you could cite the following article: Scribante et al., 2020. In Vitro Re-Hardening of Bleached Enamel Using Mineralizing Pastes: Toward Preventing Bacterial Colonization. Materials. 2020, 13, 818) but also without cytotoxicity and with specific mechanical properties.
Line 165-166: Please rephrase as “Soft polymer materials are widely used in various fields, such as in the dental field as resilient denture liner and tissue conditioner”.
If possible, I suggest adding more references in the discussion section in order to increase the number of total citations in the article (at least 25) and to reduce the percentage of self-citations.
Author Response
Response to Reviewer 3 Comments
We are grateful to Reviewer 3 for the critical comments and useful suggestions that have helped us improve our paper. As indicated in the responses that follow, we took all these comments and suggestions into account when revising our manuscript. Please note that in our revised manuscript, all revisions are red-highlighted.
Comments from Reviewer 3
INTRODUCTION
Point 1: Line 35: actually, I don’t find “to dental filed” very clear. Did you mean “in the dental field”?
Response 1: Thank you for your suggestion. I have corrected it as “in the dental field” in the manuscript.
Point 2: Line 38: “they are may also be used” should be rephrased as “they may also be used”
Response 2: Thank you for your suggestion. I have rephrased it as your suggestion in the manuscript.
Point 3: Line 40-41: “is the one of important factor” should be rephrased as “is one of the important factors”
Response 3: Thank you for your suggestion. I have rephrased it as your suggestion in the manuscript.
Point 4: Line 44: substitute “have” with “has”
Response 4: Thank you for your suggestion. I have corrected it in the manuscript.
Point 5: Line 70: substitute “materials” with “material”
Response 5: Thank you for your suggestion. I have corrected it in the manuscript.
Point 6: Line 71-72: I think the sentence should be rephrased properly
Response 6: Thank you for your suggestion. I have rephrased it as “Our study aims to develop a safe, stable, and long-lasting dental soft polymer material using citrate ester as a plasticizer by examining the dynamic viscoelasticity of the combination of citrate ester-based plasticizer and polymers.” in the manuscript.
MATERIALS AND METHODS
Point 7: Line 86: how centigrade and humidity were assessed?
Response 7: Our laboratory where the experiments were conducted in this study has a special air-conditioning system to control the temperature and humidity to maintain a constant level.
Point 8: Line 102: substitute “The specimens before immersion were used as controls” instead of “The specimens before immersion was used as a control”
Response 8: Thank you for your suggestion. I have rephrased it as your suggestion in the manuscript.
Point 9: In the section of materials and methods you should add references as regards the methodology used
Response 9: Thank you for your comment, I add the reference number in line 104.
DISCUSSION
Point 10: I suggest you to add in your discussions that many efforts should be done in dentistry not only to propose materials with anti-infective characteristics (you could cite the following article: Scribante et al., 2020. In Vitro Re-Hardening of Bleached Enamel Using Mineralizing Pastes: Toward Preventing Bacterial Colonization. Materials. 2020, 13, 818) but also without cytotoxicity and with specific mechanical properties.
Response 10: Thank you for your suggestion. I referred to the article (ref no. 22) that you recommended and add the discussions in line 244-249.
Point 11: Line 165-166: Please rephrase as “Soft polymer materials are widely used in various fields, such as in the dental field as resilient denture liner and tissue conditioner”.
Response 11: Thank you for your suggestion. I have rephrased it as your suggestion in the manuscript.
Point 12: If possible, I suggest adding more references in the discussion section in order to increase the number of total citations in the article (at least 25) and to reduce the percentage of self-citations.
Response 12: Thank you for your suggestion. I added several articles to reduce the self-citation rate and bring the total number of references up to 25.

Round 2
Reviewer 1 Report
The improvements of the authors are enough to be considered for publication